# Development of an Indirect Enzyme-Linked Immunosorbent Assay Based on the Yeast-Expressed CO-26K-Equivalent Epitope-Containing Antigen for Detection of Serum Antibodies against Porcine Epidemic Diarrhea Virus

**DOI:** 10.3390/v15040882

**Published:** 2023-03-29

**Authors:** Xuqiong Yang, Liping Li, Xiaona Su, Jiadong Li, Jiaming Liao, Jinyi Yang, Zhili Xiao, Peng Wen, Hong Wang

**Affiliations:** 1Guangdong Provincial Key Laboratory of Food Quality and Safety, College of Food Science, South China Agricultural University, Guangzhou 510642, China; 2Guangdong Laboratory for Lingnan Modern Agriculture, Guangzhou 510642, China

**Keywords:** porcine epidemic diarrhea virus (PEDV), conservative analysis, COE epitope, *Pichia pastoris* expression, iELISA

## Abstract

Porcine epidemic diarrhea (PED) is a severe contagious intestinal disease caused by the porcine epidemic diarrhea virus (PEDV), which leads to high mortality in piglets. In this study, by analyzing a total of 53 full-length spike genes and COE domain regions of PEDVs, the conserved COE fragment of the spike protein from the dominant strain SC1402 was chosen as the target protein and expressed successfully in *Pichia pastoris* (*P. pastoris*). Furthermore, an indirect enzyme-linked immunosorbent assay (iELISA) based on the recombinant COE protein was developed for the detection of anti-PEDV antibodies in pig sera. The results showed that under the optimized conditions, the cut-off value of COE-based indirect ELISA (COE-iELISA) was determined to be 0.12. Taking the serum neutralization test as standard, the relative sensitivity of the COE-iELISA was 94.4% and specificity 92.6%. Meanwhile, no cross-reactivity to other porcine pathogens was noted with this assay. The intra-assay and inter-assay coefficients of variation were less than 7%. Moreover, 164 vaccinated serum samples test showed that overall agreement between COE-iELISA and the actual diagnosis result was up to 99.4%. More importantly, the developed iELISA exhibited a 95.08% agreement rate with the commercial ELISA kit (Kappa value = 0.88), which suggested that the expressed COE protein was an effective antigen in serologic tests and the established COE-iELISA is reliable for monitoring PEDV infection in pigs or vaccine effectiveness.

## 1. Introduction

The porcine epidemic diarrhea virus (PEDV) belongs to the *Alphacoronavirus* genus in the subfamily *Coronavirinae* of the family *Coronaviridae* [1] and was first isolated from the intestinal contents of pigs in Belgium in 1978 [2,3,4]. It can cause acute enteric disease, namely porcine epidemic diarrhea (PED), which is characterized by watery diarrhea, vomiting, or dehydration in pigs, resulting in high morbidity and mortality in neonatal piglets. In recent years, the disease has frequently broken out all over the world, including in China, Japan, and many other districts in Asia. Effective diagnostic methods are continuously carried on to avoid severe economic losses caused by PED. 

Generally, enzyme-linked immunosorbent assay (ELISA) based on specific antigen-antibody recognition is a convenient tool for the diagnosis of virus infection as well as vaccine evaluation. Likewise, several ELISAs, including indirect enzyme-linked immunosorbent assay (iELISA) [5], as well as blocking ELISA [6], have been developed for the detection of anti-PEDV antibodies [7]. In particular, iELISA, compared to blocking ELISA, is more suitable for screening a large number of samples because it only requires selecting an appropriate coating antigen but not preparing another anti-PEDV antibody as a blocking antibody [8]. It is well known that PEDV mainly contains four structural proteins, namely, spike (S), envelope (E), membrane (M), and nucleocapsid (N) [9,10]. These structural proteins have already been used as target coating proteins to establish iELISAs for detecting anti-PEDV antibodies [11]; however, researchers found that N protein or M protein-based iELISAs exhibit high cross-reactivity for other swine coronaviruses [12,13] while S protein-based iELISA is more sensitive than N protein one [11]. Meanwhile, it has been proven that the S protein has higher antigenicity than other ones, and anti-S antibodies produced in PEDV-infected pigs could persist longer than anti-N antibodies [14,15]. Moreover, because the S protein harbors more virus-neutralizing epitopes, it is also recognized as the target for neutralizing antibody induction [16,17]. Therefore, the S protein would be more expected to target a suitable region of S protein to develop an iELISA for PEDV monitoring.

The CO-26K-equivalent epitope (COE epitope) is one of the various neutralizing epitopes on the S protein of PEDV, which can induce the production of neutralizing antibodies [18] and has been widely utilized for the development of subunit vaccines to prevent viral infection [19]. The neutralizing epitope region of COE contains 139 amino acids within the S1 domain extending from amino acid (aa) 499–638 [20]. The COE protein was thought to have two potential N-linked glycosylation sites [21], and the glycosylation might be necessary for its conformation stability and bioactivity [22]. Therefore, *P. pastoris*, which provides expressed heterologous protein to the post-translational glycosylation modification and ensures correct protein folding, is potentially an ideally suitable expression system for the preparation of the COE protein.

Herein, the full-length S protein amino acid sequences of 53 PEDV strains were collected and analyzed. The conserved COE epitope of the spike protein from strain SC1402 was chosen as the target protein to be expressed in *P. pastoris*. By using the recombinant COE protein as a coating antigen, an iELISA for the detection of anti-PEDV antibodies in swine serum was established and evaluated. The results showed that the developed iELISA has high specificity and reproducibility and takes less time, which suggests potential prospects in clinical application.

## 2. Materials and Methods

### 2.1. Vectors, Cells, and Sera

The COE protein was cloned using the pTOPO-T vector (Ruibiotech, Beijing, China). *E. coli* DH5α chemically competent cell was purchased from TransGen Biotech (Beijing, China). *P. pastoris* strain GS115 and plasmid pPIC9K were preserved in our laboratory. The Vero cell (ATCC: CCL-81), the PEDV CT strain, and all of the swine serum samples were kindly supplied by Wen’s Group Academy, Wen’s Foodstuffs Group Co., Ltd., China (Yunfu, Guangdong). In this study, the standard anti-PEDV positive serum and the standard negative serum not containing any pathogen antibodies were used for the optimization and evaluation of indirect ELISA. A total of 117 unconfirmed serum samples were confirmed by SN. Moreover, 130 positive serum samples from vaccinated pigs (inactivated vaccine) and 34 negative sera collected before vaccination were used for detection of vaccinated serum samples by COE-iELISA. Positive sera from pigs infected by the porcine reproductive and respiratory syndrome virus (PRRSV), classical swine fever virus (CSFV), pseudorabies virus (PRV) and porcine deltacoronavirus (PDCoV), respectively, were used for evaluation of cross-reactivity with the COE-iELISA PEDV antigen. All of the collected serum samples were centrifuged at 2500× *g* for 5 min and stored at −20 °C.

### 2.2. Phylogenetic and Conservative Analysis

A total of 53 complete genomic sequences of PEDVs were retrieved from GenBank. Information on the virus name and origin of the isolates, the year of isolation, and the GenBank accession numbers are provided in Appendix A. Sequences of PEDV S gene China isolates were compared with other representative PEDV S gene sequences using MEGA V7.0 software. The phylogenetic tree was calculated using the neighbor-joining (NJ) method. Bootstrap values were calculated based on 1000 repeats of the alignment. The amino acid sequence homology of the S gene or domain COE of 53 strains and the amino acid differences between isolates and strain SC1402 were analyzed by Clustal W with MegAlign software. The linear B cell epitopes on domain COE of PEDV were predicted using the BepiPred-2.0 server [23]. Only conserved fragments of at least 5 amino acid residues that were predicted as potential epitopes by BepiPred-2.0 were taken into consideration in this study.

### 2.3. Construction of Recombinant pPIC9K-COE Plasmid

The COE gene was subcloned into the pPIC9K plasmid by the forward primer and the reverse primer containing the 6 × His tag and the stop codon TGA. The forward primer was 5′-CGGAATTCGTTACTTTGCCATCATTCA-3′, and the reverse primer was 5′-TTGCGGCCGCTCAATGGTGATGGTGATGATGAACGTCCGTGACACC-3′. Underlined portions represent *Eco*R I and *No*t I restriction sites, respectively. The plasmid pPIC9K-COE was then chemically transformed into a strain of competent *E. coli* cells, DH5α, and the clones were identified by PCR. The resulting recombinant expression plasmid was identified by double enzyme digestion and DNA sequence analysis. Subsequently, the recombinant expression plasmid pPIC9K-COE was linearized by *Sa*c I and inserted into the genome of *P. pastoris* GS115 by electroporation (Bio-Rad, Hercules, CA, USA). Transformed cells were selected from MD plates (2% glucose, 1.34% yeast nitrogen base (YNB), and 1.8% agarose), incubated at 30 °C for 3–5 days, and were identified by PCR.

### 2.4. Expression and Purification of the Recombinant COE Protein

The positive transformants were cultured in a 10 mL YPD medium (1% yeast extract, 2% peptone, 2% glucose), shaking overnight at 30 °C. Then, overnight culture was inoculated in BMGY medium (1% yeast extract, 2% peptone, 1.34% YNB, 1% glycerol, and 100 mM potassium phosphate (pH = 6.0)) at 30 °C until the 600 nm optical density (OD_600_) of 9–10. The cells were collected by centrifugation at 4000 rpm for 10 min and resuspended to an initial OD_600_ value of 1 in BMMY medium (1% yeast extract, 2% peptone, 100 mM potassium phosphate (pH = 7.0), 1.34% YNB, and 1% methanol). Subsequently, the COE protein was induced for 96 h at 28 °C by adding methanol at a final concentration of 1% every 24 h. The culture supernatant was collected by centrifugation at 12,000 rpm for 20 min and passed through a 0.45 μm filter, then purified by Ni-NTA resin affinity chromatography using ÄKTA Pure system (GE Healthcare, Chicago, IL, USA).

### 2.5. Western Blotting 

Purified recombinant COE protein was subjected to 15% (*v*/*v*) SDS-PAGE, and the gel was prepared for western blotting as follows. Recombinant COE proteins separated in the gel were electrically transferred to a nitrocellulose membrane, and the membrane was blocked in the Tris-buffered saline containing Tween-20 (TBST, 20 mM Tris-HCl, 150 mM NaCl, and 0.05% Tween-20) which contained 5% (*w*/*v*) skimmed milk powder for 2 h at 37 °C and washed 3–5 times followed by overnight incubation at 4 °C with anti-PEDV serum (1:500). After washing 3–5 times in PBST, the membrane reacted with HRP-conjugated goat anti-swine IgG (1:5000; Sigma-Aldrich, Burlington, MA, USA) at 37 °C for 1 h. After 3–5 washes with PBST, the final color reaction was developed with a solution of 3, 3′-Diaminobenzidine (DAB).

### 2.6. Confirmation of the Serum Samples 

A total of 117 serum samples were kindly provided by Wen’s Group Academy (Wen’s Foodstuffs Group Co., Ltd., China) and confirmed by SN. Vero cells (1 × 10^5^ cells per well) were inoculated in a 96-well cell culture plate and cultured with 5% CO_2_ at 37 °C overnight until a confluent monolayer formed. Serum samples were heat-inactivated at 56 °C for 30 min, then filtered with a 0.22 μm membrane (Millipore, Burlington, MA, USA). Inactivated sera were diluted two-fold and incubated with 200 TCID_50_ viruses (PEDV CT strain) at 37 °C for 1 h. After incubation, 100 μL of the serum–virus mixture was transferred from each well of the incubation plate to a 96-well cell culture plate containing the Vero cells. The plates were incubated at 37 °C with 5% CO_2_ for 5 days. The amounts of cell wells with cytopathic effect (CPE) were counted under an inverted microscope (Mshot-MF52, Guangzhou Mingmei Optoelectronic Technology Co., Ltd., Guangzhou, China) and the neutralization titers were determined. The highest dilution of serum that can protect 50% of cells from PEDV infection is considered the neutralizing antibody titer.

### 2.7. Establishment of the COE-iELISA Method

The COE protein was used as the coating antigen to establish an indirect ELISA. ELISA plates with 96 wells were coated with 100 μL purified COE protein in coating buffer (50 mM Tris buffer, pH 8.0) for 12 h at 4 °C. Then, plates were washed twice with washing buffer (135 mM NaCl, 8 mM Na2HPO4, 0.06% Tween 20) and blocked with 5% skimmed milk in phosphate-buffered solution (10 mM PBS, pH 7.4) for 2 h at 37 °C and incubated with 100 μL of porcine serum samples diluted in PBS at 37 °C. After four washings, the plates were incubated with 100 μL HRP-conjugated goat anti-swine IgG (Sigma, Burlington, MA, USA) at 37 °C. Plates were then washed four times and were developed by the addition of 100 μL tetramethylbenzidine (TMB) substrate solution added to each well for a chromogenic reaction at room temperature for 10 min in complete darkness. Then, the reaction was stopped by adding 50 μL of 2 mol/L H_2_SO_4_. The absorbance was read at 450 nm (A_450nm_) using a microplate reader (Thermo Fisher Scientific, Waltham, MA, USA).

Based on the method described above, the optimal coating antigen concentration and standard serum dilution multiple for the COE-iELISA were determined by checkerboard titration. The additional conditions, including coating time and temperature, blocking buffer, dilution multiple of secondary antibody, and the incubation times for blocking, serum reaction, and secondary antibody, were further optimized. The factors that showed the highest ratio of positive and negative serum A_450nm_ (P_A450nm_/N_A450nm_, P/N) values were scored as optimal reaction conditions [24]. 

### 2.8. Determination of the Cut-Off Value 

The standard anti-PEDV positive and negative sera were tested in triplicate as positive and negative controls (Mean + 3 SD; OD_450nm_ ≥ 0.334 is positive, and OD_450nm_ ≤ 0.334 is negative). A total of 117 serum samples confirmed above were measured by COE-iELISA in triplicate, and the means (M) of A_450nm_ were obtained. The sample-to-positive (S/P) ratio was used for judgment and calculated by the formula as [25]
(1)$$S/P=\frac{{\;A}_{450\mathrm{nm}\;}\mathrm{of}\;\mathrm{sample}-{\;A}_{450\mathrm{nm}\;}\mathrm{of}\;\mathrm{negative}\;\mathrm{control}}{{\;A}_{450\mathrm{nm}\;}\mathrm{of}\;\mathrm{postive}\;\mathrm{control}-{\;A}_{450\mathrm{nm}\;}\mathrm{of}\;\mathrm{negative}\;\mathrm{control}}$$

The cut-off value was determined by receiver operator characteristic (ROC) curve analysis (MedCalc Statistical Software) with SN results as standard [26].

### 2.9. Cross-Reactivity and Repeatability Assay

Positive sera for PRRSV, CSFV, PRV, and PDCoV, respectively, were detected according to the COE-iELISA procedure. Each sample was examined with four repetitions, and the S/P ratios were calculated.

In addition, seven positive serum samples (SN titer ≥ 1:4) and one negative sample (SN titer < 1:4) were chosen for repeatability assay [27]. For intra-assay variability, each sample was tested twice in three replicates on the same day based on the same batch of COE protein. Moreover, for inter-assay variability, two batches of expressed COE protein were utilized. The results were presented as the coefficient of variation (CV), which is the ratio of the standard deviation (SD) to the mean OD 450 value (M) of each group of samples.

### 2.10. Detection of Vaccinated Serum Samples by COE-iELISA

A total of 130 positive serum samples from vaccinated pigs were confirmed to produce neutralizing antibodies by the challenge inoculation tests, and 34 negative sera before vaccination were detected. Then, the S/P ratios were calculated for judgment, and the overall agreement between COE-iELISA and the actual diagnosis result was evaluated. 

## 3. Results

### 3.1. Phylogenetic and Conservative Analysis of PEDVs

To choose a suitable region on the S protein as a coating antigen that could be relatively conservative among epidemic PEDV strains, we first collected 53 PEDV strains, including 46 PEDV epidemic strains isolated in China during 2017–2021 together with 7 classical strains and then carried out the phylogenetic analysis for their S genes. It can be found the PEDV strains were classified into two major subtypes, GI and GII, in which most of the Chinese pandemic strains during 2017–2021 belong to the GII subtype except the SH/SCGA strain, and the SC1402 strain belongs to the GI subtype (Appendix A). Subsequently, by comparing the aa sequences of S protein from the above strains and analyzing the homology, we observed that the SC1402 strain has only 92.67% homology to GII strains (Table 1). Therefore, to ensure the aa sequence conservation of selected coating antigen, it is necessary to further shorten the sequences and focus on the reported epitopes of S protein and make homogeneous analyses to compare between strains. Based on homology analysis of the COE domain, the results showed that the SC1402 strain shares above 96.39% homology with GII strains (Table 1). Additionally, to explore the aa variation of the COE domain of the SC1402 strain, the variances of the COE domain between PEDV strains were analyzed by comparing the aa sequences, and the result showed that a total of 30 aa mutations were observed in the PEDV strains (Appendix A). However, the aa mutation rates of the five B cell epitopes predicted for the SC1402 strain were less than 10%, except for 535F-L, 593G-S, 632Q-E/R/G/V (Table 2). These analyses demonstrated that the COE domain of strain SC1402 was highly conserved, and it was a suitable coating antigen for the establishment of detection methods.

### 3.2. Expression and Identification of Recombinant COE Protein

The recombinant COE protein was successfully expressed in the *P. pastoris* expression system. After purification by Ni-NTA resin affinity chromatography, the COE protein with an approximate molecular weight of 22 kDa was obtained, and its molecular weight was larger than the predicted 15.9 kDa (Figure 1). Furthermore, the purified COE protein was verified by Western blotting with anti-His mouse monoclonal antibody and anti-PEDV positive serum, respectively (Figure 1B). The purified COE protein was well recognized by anti-PEDV positive serum when using it as the primary antibody.

### 3.3. Optimization of COE-iELISA Reaction Conditions 

Different parameters influencing the reaction were optimized to establish an indirect ELISA method based on COE protein. As shown in Figure 2, using checkerboard titration, the concentration of coating COE protein with 1 µg/mL and 1:400 dilution multiple of negative and positive sera were firstly confirmed by the high P/N value. As shown in Figure 3, the optimum coating condition was 12 h at 4 °C. The best blocking solution was selected as 5% skimmed milk in PBS for 1 h. The optimal reaction times for serum and secondary antibodies with 1:10,000 dilutions were 45 min and 30 min, respectively.

### 3.4. Cut-Off Value of COE-iELISA

ROC curve analysis is the standard method to demonstrate the co-variation of sensitivity and specificity for systematically changed cut-off values [28]. In this study, the SN test was performed to divide the 117 swine serum samples into 27 negative samples (SN titer < 1:4) and 90 positive ones (SN titer ≥ 1:4). The ROC curve was plotted to determine the optimal cut-off value and analyze the diagnostic sensitivity and specificity of COE-iELISA using the S/P ratio compared with the SN test results. As shown in the ROC curve (Figure 4A), the area under the curve (AUC) with 0.969 ± 0.0169 (95% confidence interval was 0.919 to 0.992) showed the specificity and sensitivity (0.9 < AUC < 1) of COE-iELISA for anti-PEDV antibody detection. Moreover, the cut-off value was determined to be 0.12 with a relative sensitivity of 94.4% (85/90) and specificity of 92.6% (25/27) (Figure 4). Together, these results indicate that the established COE-iELISA was more accurate than the reported N protein-based iELISA, with an 8.5% false-positive rate and 12.7% false-negative rate.

### 3.5. Cross-Reactivity and Repeatability of COE-iELISA

To further evaluate the specificity of the COE-iELISA for anti-PEDV antibody, the cross-reactivity assay for four other swine viruses, including PRRSV, PRV, CSFV, and PDCoV, was performed. The S/P values of positive sera against PRRSV, PRV, CSFV, and PDCoV are significantly lower than the cut-off value, which means there was no cross-reactivity among the COE protein and antibodies against these four swine viruses (Table 3), indicating that the COE-iELISA was specific for anti-PEDV antibody.

Intra-assay and inter-assay are used to appraise the repeatability of the COE-iELISA. The intra-assay CV of 8 serum samples ranged from 1.58% to 5.92%, whereas the inter-assay CV was between 0.86% and 6.85% (Table 4). The results were both clearly below the in-study validation acceptance criteria of 15% [29], demonstrating that the COE-iELISA in this study was repeatable and stable.

### 3.6. Detection of Vaccinated Serum Samples of COE-iELISA

To validate the practicability of COE-iELISA, 130 positive sera from vaccinated pigs and 34 negative sera were collected before vaccination was detected. As shown in Table 5, 130 positive vaccinated serum samples were all determined as positive by COE-iELISA, while only one was misdiagnosed in 34 negative groups. The overall agreement between COE-iELISA and the actual result was up to 99.4% (163/164). It showed that the COE-iELISA established in this study could be used in anti-PEDV antibody detection in vaccinated pigs. 

### 3.7. Clinical Application

A total of 244 clinical pig serums were analyzed using the developed COE-iELISA and the commercial ELISA kit to assess the clinical application value. The developed COE-iELISA detection results revealed that among these serum samples, 171 were PEDV antibody-positive serum samples and 73 were PEDV antibody-negative serum samples, while the commercial ELISA kit showed that 173 were PEDV antibody-positive serum samples and 65 were PEDV antibody-negative serum samples. In general, among the 244 serum samples detected, 232 serum samples showed consistent results by the two methods (168+/64−), and the agreement rate was 95.08% (Kappa  =  0.88). According to the statistical analysis results, the developed COE-iELISA exhibited high agreement with the commercial ELISA kit, and there was no significant difference between the two kits (Kappa  >  0.4) (Table 6). Taken together, the developed COE-iELISA has good sensitivity and specificity and has a high agreement with commercial ELISA kits, which can be used for clinical detection.

## 4. Discussion

In China, PEDV highly pathogenic outbreak in 2010, it has caused serious economic losses to the breeding industry. Currently, PED remains a critical threat to the global swine industry [30]. Among the PEDV structural proteins, the S protein is the main envelope glycoprotein located at the surface of the virus and a surface antigen that plays an important role in interacting with the glycoprotein receptor on the host cell during infection and in mediating viral entry, inducing neutralizing antibodies and viral virulence in vivo [18,31]. Nevertheless, the detection methods targeting the S protein of PEDV lack accuracy because mutations occur in the S protein [10]. Additionally, the previous study has demonstrated that the ELISAs had not a stable and consistently higher detection based on the GI and GII strain antigens [32], and disease control still relies on vaccine immunization and the implementation of strict sanitary procedures. Thus, it is particularly important to establish a fast and efficient detection method for PEDV. 

Studies have reported that the amino acid variations of the PEDV S protein might result in antigenicity differences [15,33]. As a consequence, the selection of conserved antigen regions is very important for total antibody level and neutralizing antibody detection. In this study, by analyzing the conservation of 53 PEDV strains full-length spike genes and COE domain regions, we demonstrated that the COE domain sequences of the S protein from the SC1402 strain were highly conserved and a suitable candidate antigen for the detection of PEDV antibodies.

ELISA is suitable for large-scale clinical and epidemiological investigations and monitoring of the changes in antibody levels in pigs, which can be used to evaluate the immune efficacy after vaccination [34]. In this study, the developed iELISA based on the conserved COE domain of the S protein exhibits some unique advantages, such as a simple production process and promising eucaryotic systems. The yeast expression system could offer relatively easy genetic manipulation and achieve a high yield of protein production by shaking flask culture. In this study, the COE protein with an approximate molecular weight of 22 kDa was obtained through protein expression, and its molecular weight was larger than the predicted 15.9 kDa (Figure 1). The main reason might be that the glycosylation induced the molecular weight change. These phenomena also appeared when COE protein was expressed by through transgenic tobacco plants [35] or potato plants [36]. Secondly, the COE-iELISA could be achieved diagnosis within 2.5 h, faster than reported N protein-based iELISA with at least 4 h and M protein or S1 protein-based iELISA with over 3 h [37,38,39]. Moreover, to ensure the accuracy of the COE-iELISA, this research employed seroneutralization assay results as the standard and accurately determined the critical value through the ROC curve, and used the S/P values rather than the A_450_ of the samples directly as the judgment value of the diagnostic result, which can effectively deduct the background value of the serum to be tested and can effectively avoid the deviation of A_450_ of the test results caused by factitious operation error, making the test results more objective and credible. Notably, the sensitivity (94.4%) and specificity (92.6%) of the COE-iELISA are significantly higher than that of the indirect ELISA established by Liu et al. (sensitivity and specificity of 92.6 and 90.1%) [40]. With the continuous development of vaccines in recent years, the prevalence of TGEV and its variant PRCV in diarrhea pigs in China is very low [41], and the detection rate in clinical samples is approximately <3% [42], making it less easy to obtain positive serum samples. A previous study tested a total of 2987 clinical samples from five provinces in southern China from 2012 to 2018, and the results indicated that the rate of mono-infection of TGEV in samples tested was just 0.33% (10/2987) [43]. Thus, the inadequacies of this study lack cross-reactivity assays for TGEV and PRCV porcine coronaviruses. Nonetheless, cross-reactivity assays were conducted for four other swine viruses, including PRRSV, PRV, CSFV, and PDCoV. The cross-reactivity assay showed that the COE protein did not cross-reactivity with other viral antibodies. The detection of 130 positive vaccinated serum samples was all determined as positive by COE-iELISA, while only one was misdiagnosed in 34 negative groups. The overall agreement between COE-iELISA and the actual result was up to 99.4% (163/164). The developed iELISA exhibited a 95.08% agreement rate with the imported commercially available ELISA kit (Kappa value = 0.88). In summary, the COE-iELISA established in this study is more suitable for rapid diagnosis in large-scale sample detection and can be used in PEDV antibody detection in vaccinated pigs.

## 5. Conclusions

In the present study, we demonstrated by phylogenetic and conservative analysis that the COE domain of strain SC1402 was highly conserved among different PEDV isolates. The COE protein was successfully expressed in *P. pastoris* at a moderate level with good antigenic reactivity. The COE was used as the coating antigen to develop an indirect ELISA. The COE-iELISA showed a good level of sensitivity and specificity. The test has the potential for the detection of PEDV antibodies and is reliable for monitoring PEDV infection in pigs or vaccine effectiveness. This method has the potential to be developed into a commercial ELISA kit, which may provide new options for the prevention and control of PEDV.

## Figures and Tables

**Figure 1 viruses-15-00882-f001:**
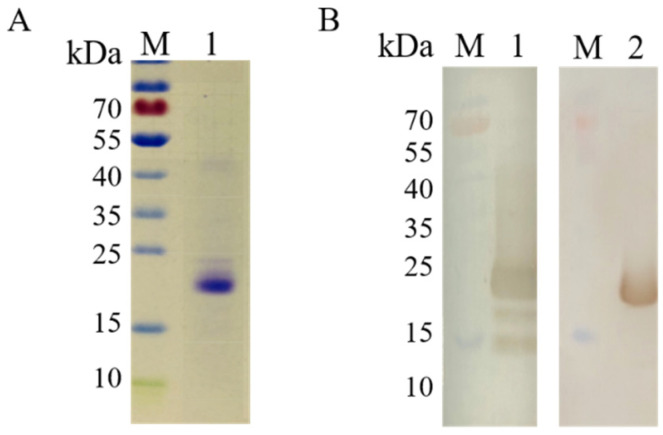
Identification of rCOE protein by SDS-PAGE and Western blotting. (**A**) Lane M, Prestained protein marker. Lane 1, purified rCOE protein. (**B**) Lane 1, Immunoblotting with anti-His mouse monoclonal antibody. Lane 2, Immunoblotting with anti-PEDV positive serum. A prominent band with the expected size of 22 kDa appeared after incubation.

**Figure 2 viruses-15-00882-f002:**
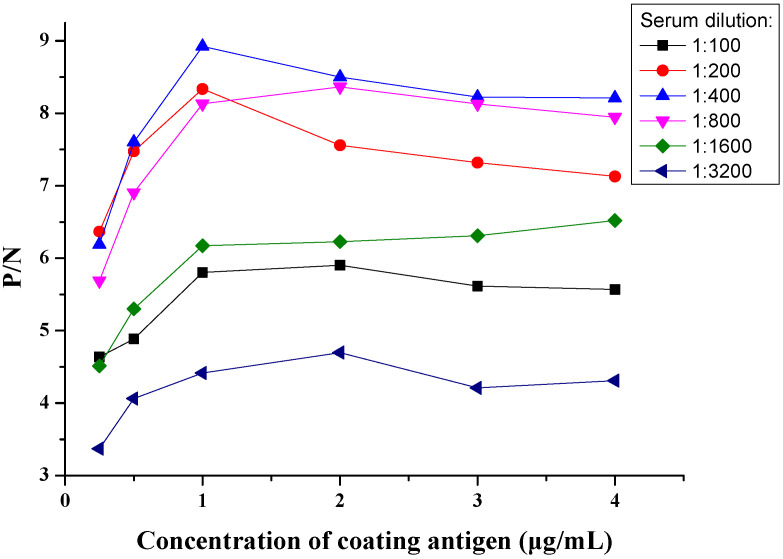
Result of checkerboard titration.

**Figure 3 viruses-15-00882-f003:**
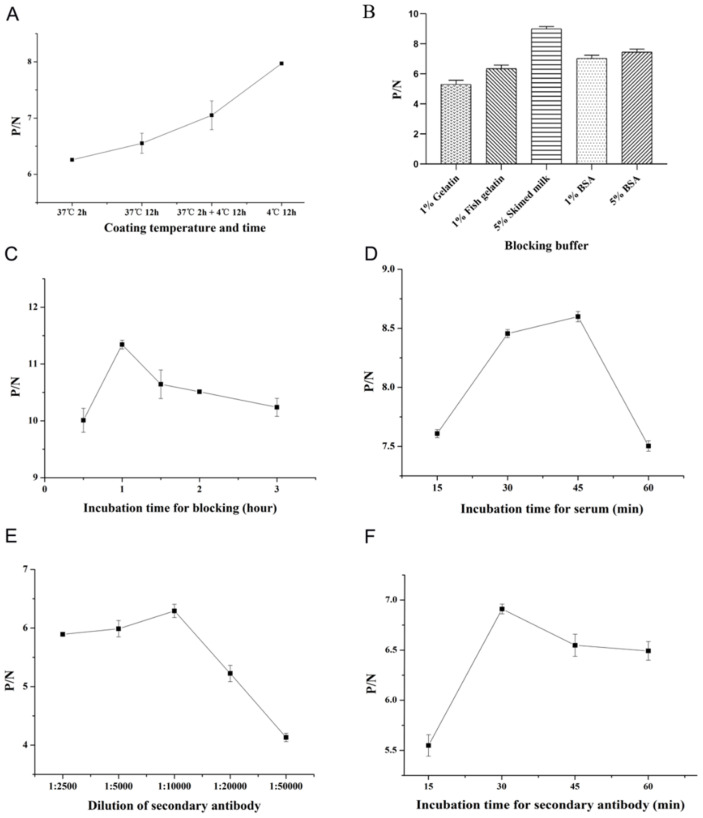
Optimization of the conditions for COE-iELISA. (**A**) COE-iELISA was optimized by different coating temperatures and times. (**B**) COE-iELISA was optimized by different blocking buffers. (**C**) COE-iELISA was optimized by different incubation times for blocking. (**D**) COE-iELISA was optimized by different incubation times for serum. (**E**) COE-iELISA was optimized by different dilutions of secondary antibodies. (**F**) COE-iELISA was optimized by different incubation times for the secondary antibody.

**Figure 4 viruses-15-00882-f004:**
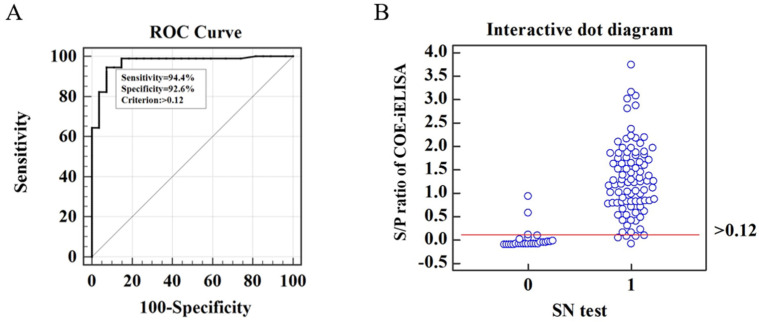
Receiver operating characteristic (ROC) curve analysis. (**A**) ROC curve of COE-iELISA with the cumulative data of the SN test as standard. (**B**) Interactive dot diagram of COE-iELISA and SN test (0 = negative samples, 1 = positive samples).

**Table 1 viruses-15-00882-t001:** Pairwise comparison of the amino acid sequences of S protein or its domain COE from GI, GII genogroup with that of PEDV reference strain SC1402 (GenBank ID: KP162057.1).

Query: SC1402	S Protein	Domain COE
GI (*n* = 7)	98.06 ± 1.96	98.16 ± 2.33
GⅡ (*n* = 45)	92.67 ± 0.89	96.39 ± 0.71

The percent amino acid identity (%) was presented as the mean ± standard deviation (SD).

**Table 2 viruses-15-00882-t002:** Statistics of mutations in the domain COE of the PEDV strains in comparison with strain SC1402.

Mutations	Mutation Rate (%)	Mutations	Mutation Rate (%)
^500^L → M/P	3.85	^589^S → N	1.92
^516^A → S/T	75.00	^590^L → P	3.84
^519^G → D	1.92	**^593^G → S**	**90.38**
^520^H → L/S/Y/P/R	21.15	^600^F → L	1.92
^521^S → I	1.92	**^604^E → A/D**	**7.69**
^522^G → S	1.92	**^607^S → G**	**5.77**
^525^L → I	9.62	**^608^G → S**	**1.92**
^526^I → V	3.84	^611^F → L	5.77
^527^A → V	1.92	^612^T → V	1.92
**^535^F → L**	**19.23**	**^627^T → M**	**1.92**
^547^I → T	1.92	**^629^K → T**	**5.77**
^548^T → S	86.54	**^631^L → F**	**1.92**
**^562^K → N/T**	**9.62**	**^632^Q → E/R/G/V**	**94.23**
**^563^S → K**	**1.92**	**^634^V → I**	**1.92**
**^565^D → H**	**7.69**	**^635^T → I**	**1.92**

The amino acid mutation rates of the five B cell epitopes predicted for the strain SC1402 are shown in bold.

**Table 3 viruses-15-00882-t003:** Specificity of the COE-iELISA to antibodies against four other swine viruses.

Virus Anti-Serum	M ± SD	Test Value (S/P)	Result
PEDV	1.279 ± 0.025	1	+
PRRSV	0.185 ± 0.014	0.016	−
PRV	0.094 ± 0.010	−0.066	−
CSFV	0.174 ± 0.008	0.006	−
PDCoV	0.183 ± 0.011	0.014	−
Negative serum	0.167 ± 0.016	0	−

A plus sign means the test result was positive, and a minus sign means the test result was negative.

**Table 4 viruses-15-00882-t004:** Intra-assay and Inter-assay repeatability of COE-iELISA.

Sample No.	Intra-Assay		Inter-Assay	
	M ± SD	CV (%)	M ± SD	CV (%)
1	0.089 ± 0.003	3.26	0.094 ± 0.008	6.59
2	0.232 ± 0.014	5.92	0.213 ± 0.015	6.85
3	0.474 ± 0.020	4.24	0.487 ± 0.034	7.01
4	0.773 ± 0.038	4.96	0.733 ± 0.013	1.74
5	1.317 ± 0.030	2.30	1.342 ± 0.038	2.87
6	1.792 ± 0.028	1.58	1.767 ± 0.015	0.86
7	2.026 ± 0.047	2.30	2.076 ± 0.069	3.33
8	2.774 ± 0.105	3.78	2.641 ± 0.060	2.27

**Table 5 viruses-15-00882-t005:** Detection results of 130 vaccinated and 34 negative serum samples by COE-iELISA.

Samples	Number	COE-iELISA	
		+	−
Vaccinated serum	130 (+)	130	0
	34 (−)	1	33

**Table 6 viruses-15-00882-t006:** Comparisons of the developed COE-iELISA with commercial ELISA kit by detecting clinical pig serum samples.

Samples	COE-iELISA	Number	Commercial ELISA Kit		Agreement (%)	Kappa Value
			+	−		
Clinical sera	+	171 (A)	168 (B)	3	95.08%	0.88
	−	73 (C)	9	64 (D)		

Agreement (%) = (B + D)/(A + C). The Kappa value > 0.4 was regarded as significant difference.

## Data Availability

Not applicable.

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
