# Peer review of "Development of an Indirect Enzyme-Linked Immunosorbent Assay Based on the Yeast-Expressed CO-26K-Equivalent Epitope-Containing Antigen for Detection of Serum Antibodies against Porcine Epidemic Diarrhea Virus"

_viruses, 2023, doi:10.3390/v15040882_

Round 1

Reviewer 1 Report

Major comments

 1. As authors mentioned that ELISA based on N protein and M protein showed high cross-reactivity with other porcine coronaviruses when detection. But in evaluation of  the specificity of the COE-iELISA for anti-PEDV antibody, why the positive serum samples for other swine coronaviruses including swine infectious gastroenteritis virus, swine respiratory coronavirus, and swine delta coronavirus were not included.

2. Line 248: It stated that the whole assay could be completed within 2 h, but according on the optimal P/N value as the screening condition, the results were blocked for 1h, serum reaction for 45min, secondary antibody for 30min, and TMB substrate for 10min. The total time is 2.35min, which is greater than 2 hours.

3 In what range of the OD values positive and negative controls is validated in the assay. Otherwise, due to the low cut-off value of this assay, the sample detected as false positive could occur when the OD value of the positive control is relatively low.

4. Low number of negative serum samples used for determining the cut-off value may lead to the inaccurate determination of the critical value, subsequently the inaccurate testing results could occur. More negative serum samples should be included in the study.

Minor comments

Line 59: “53 kinds of PEDVs” should be changed to “53 PEDV strains” were collected and analyzed.

Line 133 Which the PEDV isolate was used in SN test?

Line 138: How the neutralization titers were determined?

Line 178: What did “challenge test” mean?

Line 185 “53 kinds of PEDV strains” should be changed to “53 PEDV strains

Figure 1 could be as supplementary figure 1s

Line 231: The quality of Figure 2B should be further improved.

Line 239:  how the P/N value was calculated?

Line 259: “before vaccination was detected” should be changed to “before vaccination were detected”.

Line239: “164 vaccinated serum samples " should be changed to" 130 vaccinated and 34 negative serum samples".

Author Response

Response to Reviewer 1 Comments

  1. As authors mentioned that ELISA based on N protein and M protein showed high cross-reactivity with other porcine coronaviruses when detection. But in evaluation of the specificity of the COE-iELISA for anti-PEDV antibody, why the positive serum samples for other swine coronaviruses including swine infectious gastroenteritis virus, swine respiratory coronavirus, and swine delta coronavirus were not included.

Response 1: Thank you very much for your constructive suggestion. Due to the limitations of experimental materials, we supplemented the cross-reactivity assay for PDCoV. According to relevant reports (see references no. 1 and 2 below), the prevalence of TGEV and its variant PRCV in diarrhea pigs in China is very low, and the detection rate in clinical samples is approximately<3%, making it less easy to obtain positive serum samples. In addition, after consulting with relevant laboratories, we learned that most laboratories do not have existing positive sera, including the Professor Guihong Zhang research group at South China Agricultural University (Guangdong Provincial Key Laboratory of Comprehensive Prevention and Control for Severe Clinical Animal Diseases, College of Veterinary Medicine, Guangzhou 510642, China.,) which currently does not have TGEV, PRCV, and PDCoV positive serum at present. Therefore, we regret not being able to fully supplement the experimental data. Significantly, the COE protein did not cross-reactivity with PDCoV antibodies.

  1. Line 248: It stated that the whole assay could be completed within 2 h, but according on the optimal P/N value as the screening condition, the results were blocked for 1h, serum reaction for 45min, secondary antibody for 30min, and TMB substrate for 10min. The total time is 2.35min, which is greater than 2 hours.

Response 2: Thank you very much for your constructive suggestion. Generally, the detection time of the ELISA kit does not include the blocked time of the ELISA plates. To compare with the reported N protein-based iELISA with at least 4 h and M protein or S1 protein-based iELISA with over 3 h (see reference no. 3, 4, and 5 below), the time of the whole assay was 2 hours and 25 minutes, including the blocking time of the ELISA plate in this study. Thank you for helping us to find out about this error. We already have corrected this error pointed out by you.

3 In what range of the OD values positive and negative controls is validated in the assay. Otherwise, due to the low cut-off value of this assay, the sample detected as false positive could occur when the OD value of the positive control is relatively low.

Response 3: Thank you very much for your constructive suggestion. In this study, the negative-positive threshold was determined to be 0.334 (Mean + 3 SD), which is close to the reported indirect ELISA detection threshold (0.35) (see reference no. 6 below). The positive control OD450 value used in this study is between 1.066-1.562, and the negative control OD450 value is between 0.1-0.136.

  1. Low number of negative serum samples used for determining the cut-off value may lead to the inaccurate determination of the critical value, subsequently the inaccurate testing results could occur. More negative serum samples should be included in the study.

Response 4: As shown in Table 5, 130 positive vaccinated serum samples were all determined as positive by COE-iELISA, while only one was misdiagnosed in 34 negative groups, which was presumably caused by the interference of serum background. It can be seen from this result that COE-iELISA is highly consistent with the actual negative sample. Further, Fan et al have determined the cut-off value of indirect ELISA with 36 negative serum samples (see reference no. 3 below). The number of negative samples was close to the number used in this study. For the above reasons, we believe that the critical value determined by the above method is reliable and accurate.

Minor comments

Line 59: “53 kinds of PEDVs” should be changed to “53 PEDV strains” were collected and analyzed.

Line 133 Which the PEDV isolate was used in SN test?

Line 138: How the neutralization titers were determined?

Line 178: What did “challenge test” mean?

Line 185 “53 kinds of PEDV strains” should be changed to “53 PEDV strains

Figure 1 could be as supplementary figure 1s

Line 231: The quality of Figure 2B should be further improved.

Line 239:  how the P/N value was calculated?

Line 259: “before vaccination was detected” should be changed to “before vaccination were detected”.

Line 239: “164 vaccinated serum samples " should be changed to" 130 vaccinated and 34 negative serum samples".

Response: Thank you very much for your constructive suggestion. We have revised our manuscript following all your above suggestions.

References:

  1. Zhang F, Luo S, Gu J, Li Z, Li K, Yuan W, Ye Y, Li H, Ding Z, Song D, Tang Y. Prevalence and phylogenetic analysis of porcine diarrhea associated viruses in southern China from 2012 to 2018. BMC Vet Res. 2019 Dec 27;15(1):470. doi: 10.1186/s12917-019-2212-2.
  2. Chen Y, Zhang Y, Wang X, Zhou J, Ma L, Li J, Yang L, Ouyang H, Yuan H, Pang D. Transmissible Gastroenteritis Virus: An Update Review and Perspective. Viruses. 2023 Jan 27;15(2):359. doi: 10.3390/v15020359.
  3. Fan, J. H.; Zuo, Y. Z.; Shen, X. Q.; Gu, W. Y.; Di, J. M., Development of an enzyme-linked immunosorbent assay for the monitoring and surveillance of antibodies to porcine epidemic diarrhea virus based on a recombinant membrane protein. J Virol Methods 2015, 225, 90-4.
  4. Li, Y.; Zheng, F.; Fan, B.; Muhammad, H. M.; Zou, Y.; Jiang, P., Development of an indirect ELISA based on a truncated S protein of the porcine epidemic diarrhea virus. Can J Microbiol 2015, 61, (11), 811-7.
  5. Okda, F.; Liu, X.; Singrey, A.; Clement, T.; Nelson, J.; Christopher-Hennings, J.; Nelson, E. A.; Lawson, S., Development of an indirect ELISA, blocking ELISA, fluorescent microsphere immunoassay and fluorescent focus neutralization assay for serologic evaluation of exposure to North American strains of Porcine Epidemic Diarrhea Virus. BMC Vet Res 2015, 11, 180.

Reviewer 2 Report

1.      Lines 196-198: “The COE domain (aa 499-638) is one of the various…”. The description is better moved to the introduction section, and more information about the domain should be included.

2.      An introduction of strain SC1402 and the reason why the authors chose this strain should be described in either the Introduction or Discussion section.

3.      Lines 72-78: more information should be provided for the sera, e.g, sources, ages, and ways to confirm the infection.

4.      In paragraph 3.3., specify the sub-figures of Figure 4 (i.e., Fig. 4A to Fig. 4F) right after the sentences that describe each of them.

5.      Figure 4B, the five blocking conditions are independent and should be displayed by a bar graph instead of a line graph.

6.      Figure 5, the subfigure captions (A and B) are shifted.

7.      The “original images for blots/gels”provided is still cropped. Please upload the three original images for the gel and the two blots.

Author Response

Response to Reviewer 2 Comments

  1. Lines 196-198: “The COE domain (aa 499-638) is one of the various…”. The description is better moved to the introduction section, and more information about the domain should be included.

Response 1: Thank you very much for your constructive suggestion. We already added this information to the Introduction section of the revised manuscript.

  1. An introduction of strain SC1402 and the reason why the authors chose this strain should be described in either the Introduction or Discussion section.

Response 2: Thank you very much for your constructive suggestion. We already added this information to the Discussion section of the revised manuscript.

  1. Lines 72-78: more information should be provided for the sera, e.g, sources, ages, and ways to confirm the infection.

Response 3: Thank you very much for your constructive suggestion. The sera involved in the article has been donated by Wen's Group Academy, Wen's Foodstuffs Group Co., Ltd., China., and the specific information has been written in the materials and methods.

  1. In paragraph 3.3., specify the sub-figures of Figure 4 (i.e., Fig. 4A to Fig. 4F) right after the sentences that describe each of them.

Response 4: Thank you very much for your constructive suggestion. We already added this information to our resubmitted revised manuscript.

  1. Figure 4B, the five blocking conditions are independent and should be displayed by a bar graph instead of a line graph.

Response 5: Thank you for helping us to find out this error. We already have corrected this error pointed out by you.

  1. Figure 5, the subfigure captions (A and B) are shifted.

Response 6: Thank you for helping us to find out this error. We already have corrected this error pointed out by you.

  1. The “original images for blots/gels” provided is still cropped. Please upload the three original images for the gel and the two blots.

Response 7: Thank you for helping us to find out this error. We already have corrected this error pointed out by you.

Reviewer 3 Report

In the manuscript entitled: “Development of an indirect enzyme-linked immunosorbent as-2 say to detect PEDV serum antibodies using the COE protein ex-3 pressed in Pichia pastoris”, the authors established an indirect ELISA method to detect antibodies against PEDV by using the conserved COE fragment of the S protein of the dominant PEDV strain as the coating antigen. The method exhibited good sensitivity, specificity and repeatability. The experimental design and results are adequate, however, there still remain several inadequate points or misunderstandings throughout the paper.

Major comments:

Comment 1: In this study, only serum antibodies was tested by the established method. However, in clinical practice, the IgA antibodies in the colostrum of the sows are considered to be crucial for protecting the neonatal piglets from PEDV infection. Milk samples of vaccinated sows should be tested by the method.  

Comment 2: If the method goes to application in the future, more information should be provided. What is the valid period of the coated plates under different storage temperature? What is the price of each sample?

Comment 3: In the results part, the sentences that analyze and explain the result data should be moved to the discussion part, and the structure should be re-organized.

Minor comments:

Comment 1: The quality of figure 2B is poor, the protein mark could hardly be seen, please replace it.

Comment 2: line 226-227, the purity and the yield of the protein cannot be calculated based on the western-blot figure, please add related experiment data.

Comment 3: The language of the paper should be further polished. Line 239 “highest”, line 327 “always”, line 328 “none of”, line 356 “no” are inappropriate in a scientific paper.

Author Response

Response to Reviewer 3 Comment

Comment 1: In this study, only serum antibodies were tested by the established method. However, in clinical practice, the IgA antibodies in the colostrum of the sows are considered to be crucial for protecting the neonatal piglets from PEDV infection. Milk samples of vaccinated sows should be tested by the method.

Response 1: Thank you very much for your constructive suggestion. As the reviewer suggested, maternal antibodies have a protective effect on piglets against PEDV. The higher the IgA content in colostrum, the stronger the resistance of piglets to PEDV infection. Therefore, IgA in the colostrum of the sow is more suitable as a molecular marker to evaluate the protection rate (see reference no. 1 below ). However, the purpose of this study was to establish an anti-PEDV antibodies detection method with high detection performance, economic and rapid production, and simple operation. This method can be used for clinical PEDV diagnosis and antibody monitoring, and provide technical means for PEDV epidemiological investigation, prevention and control, and PEDV vaccine evaluation.

Comment 2: If the method goes to application in the future, more information should be provided. What is the valid period of the coated plates under different storage temperature? What is the price of each sample?

Response 2: Thank you very much for your constructive suggestion. The focus of this experiment was to explore the effectiveness of the establishment method, and the work suggestions put forward by the reviewer will be carried out in the later practical application.

Comment 3: In the results part, the sentences that analyze and explain the result data should be moved to the discussion part, and the structure should be re-organized.

Response 3: Thank you very much for your constructive suggestion. We already added this information to the discussion section of our resubmitted revised manuscript.

Minor comments:

Comment 1: The quality of figure 2B is poor, the protein mark could hardly be seen, please replace it.

Comment 2: line 226-227, the purity and the yield of the protein cannot be calculated based on the western-blot figure, please add related experiment data.

Comment 3: The language of the paper should be further polished. Line 239 “highest”, line 327 “always”, line 328 “none of”, line 356 “no” are inappropriate in a scientific paper.

Response: Thank you very much for your constructive suggestion. We have revised our manuscript following all your above suggestions.

References:

  1. Ha, G.W., Hanyang University, Ansan, Republic of Korea, Kang, B.K., Research Unit, Green Cross Veterinary Products, Yongin, Republic of Korea, Park, S.J., Seoul National University, Seoul, Republic of Korea, et al. The colostral IgA level, but not the serum neutralization titer, induced by immunization against porcine epidemic diarrhea virus is correlated with protection of new born piglets against virulent PEDV challenge[J]. Korean Journal of Veterinary Public Health, 2010.

Reviewer 4 Report

This manuscript by Yang et al. describes the generation of a recombinant Pichia pastoris expressing the PEDV COE epitope and development of an indirect enzyme-linked immunosorbent assay to detect PEDV serum antibodies using the COE protein. Although spike protein has already been used as target coating proteins to establish iELISAs for detecting anti-PEDV antibodies, the novelty of this manuscript is that the conserved CO-26K-equivalent epitope (COE epitope) of the spike protein from strain SC1402 was chosen as the target protein to be expressed in P. pastorisa.

To improve the quality of the manuscript, please consider following points:

1.      In cross-reactivity experiments, PRRSV, CSFV and PRV were used; why not detect the cross-reactivity against transmissible gastroenteritis virus (TGEV), porcine respiratory coronavirus (PRCV), porcine deltacoronavirus (PDCoV) or porcine rotavirus (PoRV) pig antisera.

2.      If the aim of phylogenetic and conservative analysis was to screen the conserved region of COE to establish iELISAs for detecting anti-PEDV antibodies against GI and GII PEDV strains, the serum collected from pigs infected with GI or GII PEDV strains should be well defined.

3.    In the discussion part, the relative sensitivity and specificity of the COE-iELISA should be compared with other iELISAs for detecting anti-PEDV antibodies.

4.      Line 75, “positive serum samples from vaccinated pigs”, the name of vaccine should be listed.

5.      Line 227, “the purity of the purified COE was up to 95% and the yield was over 3.3 mg/L.” How to calculate the yield of purified COE protein?

6.      Figure 2 panel b is not clear, especially the marker. Please provide three replicates.

Author Response

Response to Reviewer 4 Comments

  1. In cross-reactivity experiments, PRRSV, CSFV and PRV were used; why not detect the cross-reactivity against transmissible gastroenteritis virus (TGEV), porcine respiratory coronavirus (PRCV), porcine deltacoronavirus (PDCoV) or porcine rotavirus (PoRV) pig antisera.

Response 1: Thank you very much for your constructive suggestion. Due to the limitations of experimental materials, we supplemented the cross-reactivity assay for PDCoV. According to relevant reports (see references no. 1 and 2 below), the prevalence of TGEV and its variant PRCV in diarrhea pigs in China is very low, and the detection rate in clinical samples is approximately<3%, making it less easy to obtain positive serum samples. In addition, after consulting with relevant laboratories, we learned that most laboratories do not have existing positive sera, including the Professor Guihong Zhang research group at South China Agricultural University (Guangdong Provincial Key Laboratory of Comprehensive Prevention and Control for Severe Clinical Animal Diseases, College of Veterinary Medicine, Guangzhou 510642, China.,) which currently does not have TGEV, PRCV, and PDCoV positive serum at present. Therefore, we regret not being able to fully supplement the experimental data. Significantly, the COE protein did not cross-reactivity with PDCoV antibodies.

  1. If the aim of phylogenetic and conservative analysis was to screen the conserved region of COE to establish iELISAs for detecting anti-PEDV antibodies against GI and GII PEDV strains, the serum collected from pigs infected with GI or GII PEDV strains should be well defined.

Response 2: Thank you very much for your constructive suggestion. Since the source of serum samples is were kindly provided by Wen's Group Academy, and the serum collected from pigs infected with GI or GII PEDV strains was no effective distinction, we did not carry out differentiation verification for GI or GII type serums. This is also the reason why we established the detection method by analyzing the highly conserved region COE as the coating antigen. Furthermore, the clinical pig serums were analyzed using the developed COE-iELISA and the commercial ELISA kit to assess the clinical application value. Among the 244 serum samples detected, 232 serum samples showed consistent results by the two methods (168+/64-), and the agreement rate was 95.08% (Kappa = 0.88). That suggested that the established COE-iELISA is reliable to monitor PEDV infection in pigs or vaccine effectiveness.

  1. In the discussion part, the relative sensitivity and specificity of the COE-iELISA should be compared with other iELISAs for detecting anti-PEDV antibodies.

Response 3: Thank you very much for your constructive suggestion. We already added this information to the discussion section of our resubmitted revised manuscript.

  1. Line 75, “positive serum samples from vaccinated pigs”, the name of vaccine should be listed.

Response 4: Thank you very much for your constructive suggestion. We have revised our manuscript following your above suggestion.

  1. Line 227, “the purity of the purified COE was up to 95% and the yield was over 3.3 mg/L.” How to calculate the yield of purified COE protein?

Response 5: Thank you very much for your constructive suggestion. The concentration of the purified COE protein was measured by Nanodrop 2000C, and then the yield of purified COE protein was calculated by the volume of the medium used for the protein expression.

  1. Figure 2 panel b is not clear, especially the marker. Please provide three replicates.

Response 6: Thank you for helping us to find out this error. We already have corrected this error pointed out by you.

References:

  1. Zhang F, Luo S, Gu J, Li Z, Li K, Yuan W, Ye Y, Li H, Ding Z, Song D, Tang Y. Prevalence and phylogenetic analysis of porcine diarrhea associated viruses in southern China from 2012 to 2018. BMC Vet Res. 2019 Dec 27;15(1):470. doi: 10.1186/s12917-019-2212-2.
  2. Chen Y, Zhang Y, Wang X, Zhou J, Ma L, Li J, Yang L, Ouyang H, Yuan H, Pang D. Transmissible Gastroenteritis Virus: An Update Review and Perspective. Viruses. 2023 Jan 27;15(2):359. doi: 10.3390/v15020359.
